

# A multicentre study to determine the *in vitro* efficacy of flomoxef against extended-spectrum beta-lactamase producing *Escherichia coli* in Malaysia

Polly Soo Xi Yap[1], Chun Wie Chong[2], Sasheela Ponnampalavanar[3], Ramliza Ramli[4], Azian Harun[5,6], Tengku Zetty Maztura Tengku Jamaluddin[7], Anis Ahmed Khan[8], Soo Tein Ngoi[9], Yee Qing Lee[10], Min Yi Lau[10], Shiang Chiet Tan[10], Zhi Xian Kong[10], Jia Jie Woon[10], Siew Thong Mak[10], Kartini Abdul Jabar[10], Rina Karunakaran[10], Zalina Ismail[4], Sharifah Azura Salleh[4], Siti Suraiya Md Noor[5], Siti Norbaya Masri[7], Niazlin Mohd Taib[7], Azmiza Syawani Jasni[7], Loong Hua Tee[11], Kin Chong Leong[11], Victor Kok Eow Lim[8], Sazaly Abu Bakar[12] and Cindy Shuan Ju Teh[10]

[1] Jeffrey Cheah School of Medicine and Health Science, Monash University Malaysia, Bandar Sunway, Selangor, Malaysia
[2] School of Pharmacy, Monash University Malaysia, Bandar Sunway, Selangor, Malaysia
[3] Department of Medicine, Faculty of Medicine, Universiti Malaya, Kuala Lumpur, Malaysia
[4] Department of Medical Microbiology and Immunology, Faculty of Medicine, Universiti Kebangsaan Malaysia, Kuala Lumpur, Kuala Lumpur, Malaysia
[5] Department of Medical Microbiology and Parasitology, School of Medical Sciences, Universiti Sains Malaysia, Health Campus, Kubang Kerian, Kelantan, Malaysia
[6] Hospital Universiti Sains Malaysia, Kubang Kerian, Kelantan, Malaysia
[7] Department of Medical Microbiology and Parasitology, Faculty of Medicine and Health Sciences, Universiti Putra Malaysia, Serdang, Selangor, Malaysia
[8] School of Medicine, International Medical University, Bukit Jalil, Kuala Lumpur, Malaysia
[9] Department of Anesthesiology, Faculty of Medicine, Universiti Malaya, Kuala Lumpur, Kuala Lumpur, Malaysia
[10] Department of Medical Microbiology, Faculty of Medicine, Universiti Malaya, Kuala Lumpur, Malaysia
[11] Shionogi Singapore, Singapore, Singapore
[12] Tropical Infectious Diseases Research and Education Centre (TIDREC), Universiti Malaya, Kuala Lumpur, Malaysia

Corresponding author
Cindy Shuan Ju Teh,
cindysjteh@um.edu.my

## ABSTRACT

**Background:** The high burden of extended-spectrum beta-lactamase-producing (ESBL)-producing Enterobacterales worldwide, especially in the densely populated South East Asia poses a significant threat to the global transmission of antibiotic resistance. Molecular surveillance of ESBL-producing pathogens in this region is vital for understanding the local epidemiology, informing treatment choices, and addressing the regional and global implications of antibiotic resistance.
**Methods:** Therefore, an inventory surveillance of the ESBL-*Escherichia coli* (ESBL-EC) isolates responsible for infections in Malaysian hospitals was conducted. Additionally, the *in vitro* efficacy of flomoxef and other established antibiotics against ESBL-EC was evaluated.
**Results:** A total of 127 non-repetitive ESBL-EC strains isolated from clinical samples were collected during a multicentre study performed in five representative Malaysian

hospitals. Of all the isolates, 33.9% were isolated from surgical site infections and 85.8% were hospital-acquired infections. High rates of resistance to cefotaxime (100%), cefepime (100%), aztreonam (100%) and trimethoprim-sulfamethoxazole (100%) were observed based on the broth microdilution test. Carbapenems remained the most effective antibiotics against the ESBL-EC, followed by flomoxef. Antibiotic resistance genes were identified by PCR. The $bla_{CTX-M-1}$ was the most prevalent ESBL gene, with 28 isolates (22%) harbouring $bla_{CTX-M-1}$ only, 27 isolates (21.3%) co-harbouring $bla_{CTX-M-1}$ and $bla_{TEM}$, and ten isolates (7.9%) co-harbouring $bla_{CTX-M-1}$, $bla_{TEM}$ and $bla_{SHV}$. A generalised linear model showed significant antibacterial activity of imipenem against different types of infection. Besides carbapenems, this study also demonstrated a satisfactory antibacterial activity of flomoxef (81.9%) on ESBL-EC, regardless of the types of ESBL genes.

# INTRODUCTION

*Enterobacterales* producing extended-spectrum beta-lactamases (ESBLs) poses a health security threat that demands global attention due to the limited treatment options available (*World Health Organization (WHO), 2020*, *2021*; *Gutierrez-Gutierrez & Rodriguez-Bano, 2019*). *Escherichia coli* species was identified as the most prominent isolate among various ESBL producers, including *Klebsiella pneumoniae* and *Klebsiella oxytoca* (*Eltai et al., 2018*; *Gharavi et al., 2021*). ESBL-producing *E. coli* (ESBL-EC) have been associated more frequently with urinary tract infections (*Centers for Disease Control and Prevention, 2019*; *Eltai et al., 2018*; *Gharavi et al., 2021*) and colonisation related to intensive care unit (ICU) admission (*Liu et al., 2018*; *Repesse et al., 2017*). Furthermore, surgical site infections (SSIs) caused by ESBL-producers have been increasingly reported, causing a considerable burden to patients and the healthcare system (*Jolivet et al., 2018*; *Kalakouti et al., 2017*).

ESBLs typically confer resistance to penicillins and cephalosporins. Frequent co-resistance to non-beta-lactam antibiotics such as sulphonamides and fluoroquinolones has also been reported, thus limiting the availability of other therapeutic options (*Gutierrez-Gutierrez & Rodriguez-Bano, 2019*). Carbapenems have been widely used as the treatment choice for ESBL-EC infections. However, the dramatic global emergence of carbapenem-resistant *Enterobacterales* (CRE) threatens the efficacy of this class of antibiotics (*World Health Organization (WHO), 2020*). These unwelcome trends have spurred the interest in exploring carbapenem-sparing alternatives. For instance, repurposing fosfomycin has gained renewed interest. Still, its safety and efficacy for treating severe infections remained a major concern, (*Sojo-Dorado et al., 2022*). A randomised controlled trial investigating antibiotic rotation between third- or fourth-generation cephalosporins, piperacillin-tazobactam and carbapenems in eight ICUs showed that antibiotic cycling does not reduce the prevalence of ICU-carriage of Gram-negative antibiotic-resistant bacteria (*van Duijn et al., 2022*). Strategies for

carbapenem de-escalation in severe infections involving ESBL-producing *Enterobacterales* were also evaluated (*Lew et al., 2015*; *Sadyrbaeva-Dolgova et al., 2019*).

Malaysia, one of the low- and middle-income countries (LMICs) in Southeast Asia (SEA), is not spared from being confronted with a significant burden of antimicrobial resistance (AMR) (*Naeemmudeen et al., 2021*). Among the effort to minimise the threat of AMR, the Malaysian government has introduced the action plan, namely MyAP-AMR (2017–2021), which incorporated multiple sectors concerning human and animal health. One of the antimicrobial stewardship programmes implemented was the "carbapenem 72-hour stop order", which required review and justification on the prescription of carbapenem beyond 72 h by the clinicians (*Ministry of Health Malaysia, 2017*). As demonstrated by Malaysian the National Antibiotic Resistance Surveillance Report (NSAR), the carbapenem resistance rates among *E. coli* isolates are still less than 1% for the past 5 years (*Institute of Medical Research (IMR), 2020*). Published data which is clinically orientated and focused on ESBL-EC are scarce; the few studies published concerning prevalence or risk factors of such infections/colonisation involved only small samples of isolates and were carried out in individual centres (*Abubakar et al., 2022*; *Dwiyanto et al., 2020*; *Ngoi et al., 2021*).

Flomoxef, a broad-spectrum beta-lactam antibiotic, was first synthesised in Japan in the 1980s. It is active against both Gram-negative and Gram-positive organisms (*Ito & Ishigami, 1991*; *Tsuji et al., 1985*). Flomoxef exhibits several properties that make it a potentially promising agent for alternative treatment in LMICs where ESBLs infections and colonisation are prevalent. Early clinical studies have confirmed its high stability to degradation by ESBLs (except for AmpC beta-lactamases) and favourable safety profile (*Ito & Ishigami, 1991*; *Lee et al., 2007*; *Matsumura et al., 2016*). Recent reports from *Darlow & Hope (2021)* and *Darlow et al. (2022)* have assessed the potential utility and safety of flomoxef in treating neonatal sepsis caused by ESBL-producing *Enterobacterales*. Furthermore, flomoxef is off-patent, favouring its local affordability in LMICs settings (*Ito & Ishigami, 1991*).

In this multicentre study, we examined the *in vitro* activity of flomoxef together with other frequently used antibiotics against clinically isolated ESBL-EC. This is an important study exploring an alternative agent with increased promising use in the 'post-antibiotic' era with high ESBLs prevalence in both clinically and community settings. Additionally, molecular characteristics of ESBLs may exhibit substantial local variations, multicentre study is critical to gain comprehensive knowledge of these isolates to allow monitoring trends of AMR and antibiotic usage. Data obtained from this study can inform the development of treatment guidelines and the implementation of infection prevention and AMR control programmes.

## MATERIALS AND METHODS

### Study site, patient data collection and definitions

The study was conducted at four tertiary teaching hospitals (UMMC; University Malaya Medical Centre, HCTM UKM; Hospital Canselor Tuanku Muhriz University Kebangsaan Malaysia, HS; Hospital Serdang and HUSM; Hospital Universiti Sains Malaysia) and HK-

IMU; Hospital Kluang-International Medical University, a district hospital in Johor. Convenient sampling method was adopted in this study. Inclusion criteria were isolates that were related to post-surgical and deep-seated infections (including tissues, bone, pus) or isolates from sterile sites including tissues, bone and blood. Isolates from swabs or non-sterile sites and those without clinical and patients' demography data were excluded. The isolates were collected between June 2017 and January 2020. Patient demographic and clinical data were retrieved from the hospitals' medical record unit. Patients' age, ethnicity, gender, admission diagnosis, type of infection (TOI; SSI and non-SSI), and infection acquired model (IAM; hospital-acquired and community-acquired) were tabulated for further analysis. An SSI is an infection that occurs postoperatively in the part of the body where the surgery took place (*Ducel, Fabry & Nicolle, 2002*). The remaining infections were categorised as non-SSI. Hospital-acquired infections are infections that occur more than 48 h after admission (*Ducel, Fabry & Nicolle, 2002*). The remaining infections were categorised as community-acquired infections. The study was conducted according to the guidelines of the Declaration of Helsinki. The study was registered in the Malaysian National Medical Research Register (NMRR) and approved by the Malaysian Medical Research and Ethics Committee (MREC) (NMRR-19-1314-46718 IIR).

## Bacterial strains

A total of 127 ESBL-EC were recovered from the hospital diagnostic laboratories. The isolation and identification of the bacterial strains and the detection of ESBL production were part of the routine microbiological examination procedures in the hospitals' diagnostic laboratories. The bacterial isolates were transported in inoculated Luria-Bertani agar stab cultures at ambient temperature. All bacterial strains were submitted to the central laboratory in the University Malaya for species re-confirmation and antimicrobial susceptibility testing. All bacterial isolates collected from the five hospitals were subjected to disk diffusion procedure using ceftazidime (30 μg), cefotaxime (30 μg), cefotaxime/clavulanate (30/10 μg), and ceftazidime/clavulanate (30/10 μg) for the confirmation of ESBL production as recommended by the Clinical and Laboratory Standards Institute (CLSI) M100 protocol (*Clinical & Laboratory Standards Institute (CLSI), 2020*). Only the first isolate of ESBL-producer from patient was included in this study.

## Antibiotic susceptibility testing

*In vitro* susceptibility of additional agents such as cefpodoxime (10 μg), ceftriaxone (30 μg), ciprofloxacin (5 μg) and ertapenem (10 μg) was performed by using the disk diffusion method at the central laboratory. The minimum inhibitory concentrations (MICs) of flomoxef and other cephems (cefoxitin, ceftazidime, cefotaxime, and cefepime); monobactam (aztreonam); beta-lactam combination agents (amoxicillin-clavulanate and piperacillin-tazobactam); carbapenems (imipenem and meropenem); and trimethoprim-sulfamethoxazole against ESBL-EC were determined using broth microdilution method. All testing was performed in accordance to the CLSI M100 protocol, and the MIC breakpoints were determined based on the recommended

guidelines (*Clinical & Laboratory Standards Institute (CLSI), 2020*). Since flomoxef has been mainly used in East Asia (*Jung et al., 2019*), the CLSI-recommended breakpoints for flomoxef were unavailable. The MIC breakpoints for moxalactam were used instead for comparison purposes as previously done in other studies (*Jung et al., 2019*; *Ngoi et al., 2021*; *Yang et al., 2015*).

### Detection of beta-lactamase-encoding genes

PCR was performed to detect the presence of ESBL genes (blaCTX-M-1, blaTEM and blaSHV), the plasmid-borne AmpC-type beta-lactamase gene (blaFOX) and carbapenemase genes (blaOXA-48, blaNDM, blaVIM, blaIMP, and blaKPC) among the ESBL-positive strains. Primer sequences, PCR reaction mixture, and thermocycling conditions were adapted from published studies (*Dallenne et al., 2010*; *Eckert et al., 2004*; *Ngoi et al., 2021*). Primers information was also included in Table S1.

### Statistical analyses

Principal coordinate analysis (PCoA) and generalised linear model (GLM) were performed to compare the space distribution difference and to identify possible effects of the TOI and IAM (explanatory variables) on the isolates' susceptibility phenotypes (response variables). Subsequently, a Pearson's Chi-squared test was conducted to evaluate the association of TOI/IAM with the demographic factors. All data analyses were carried out using R version 4.0.4 (*R Core Team, 2021*).

## RESULTS

### Distribution of bacteria and antibiotic susceptibility

During the study period, a total of 127 ESBL-EC was isolated from UMMC ($n = 41$), HCTM UKM ($n = 32$), HUSM ($n = 40$), HS ($n = 7$) and Hospital Kluang ($n = 7$) (Table 1). A total of 59.8% of the ESBL-EC were isolated from Malay patients, followed by Chinese (19.7%) and Indian (14.2%). Seventy-six (59.8%) isolates were from male patients. Majority of the isolates (85.8%) were associated with nosocomial infections. On the other hand, 33.9% of the isolates were associated with SSI, and 66.1% were non-SSI.

Based on the disk diffusion test conducted at the central laboratory, the isolates were highly resistant to both cephalosporins, cefpodoxime (99.2%) and ceftriaxone (99.2%) while 57.5% of the isolates were resistant to ciprofloxacin. All isolates were susceptible to ertapenem. The distribution of the minimum inhibitory concentration (MIC) range, median ($MIC_{50}$) and 90% efficacy ($MIC_{90}$) values of all antibiotics tested as well as the corresponding phenotypes of the ESBL-EC strains are presented in Table 2. High MIC values ($MIC_{50} > 256$ µg/mL) were recorded for monobactam and third- and fourth-generation cephalosporins. Seventy-six percent of the isolates were resistant to the beta-lactam combination agent, piperacillin-tazobactam with a $MIC_{50}$ value of >512 µg/mL. A hundred percent of the isolates were resistant to cefotaxime, cefepime, aztreonam and trimethoprim-sulfamethoxazole. The majority of the isolates remained sensitive to carbapenems, including imipenem (81.9%) and meropenem (97.6%). Flomoxef showed a $MIC_{50}$ value of 4 µg/mL, and 81.9% of the isolates remained susceptible to this

**Table 1 ESBL-EC isolate information.** Source breakdown of 127 ESBL-EC isolates by year, hospital type, region, patient demography, type of infection and infection acquired model.

| Source category | Number (% of source category) of isolates from each year | | | |
|---|---|---|---|---|
| | **2017** | **2018** | **2019** | **Total** |
| Total | 31 | 25 | 71[*] | 127 |
| Hospital type | | | | |
| Tertiary teaching hospital | 31 (100) | 25 (100) | 64[*] (90.1) | 120 (94.5) |
| District hospital | 0 (0) | 0 (0) | 7 (9.9) | 7 (5.5) |
| Region of Peninsular Malaysia | | | | |
| Northeastern | 2 (6.5) | 11 (44.0) | 26 (36.6) | 39 (30.7) |
| Central | 29 (93.5) | 14 (56.0) | 38[*] (53.5) | 81 (63.8) |
| Southern | 0 (0) | 0 (0) | 7 (9.9) | 7 (5.5) |
| Gender | | | | |
| Male | 21 (16.5) | 13 (10.2) | 42 (33.1) | 76 (59.8) |
| Female | 10 (7.9) | 12 (9.4) | 29[*] (22.8) | 51 (40.2) |
| Ethnicity | | | | |
| Malay | 9 (7.1) | 13 (10.2) | 54[*] (42.5) | 76 (59.8) |
| Chinese | 10 (7.9) | 8 (6.3) | 7 (5.5) | 25 (19.7) |
| India | 9 (7.1) | 3 (2.4) | 6 (4.7) | 18 (14.2) |
| Others | 3 (2.4) | 1 (0.8) | 4 (3.1) | 8 (6.3) |
| Type of infection (TOI) | | | | |
| SSI | 12 (38.7) | 13 (52) | 18[*] (25.4) | 43 (33.9) |
| Non-SSI | 19 (61.3) | 12 (48) | 53 (74.6) | 84 (66.1) |
| Infection acquired model (IAM) | | | | |
| Hospital-acquired | 27 (87.0) | 23 (92.0) | 59[*] (83.1) | 108 (85.8) |
| Community-acquired | 4 (13.0) | 2 (8.0) | 12 (16.9) | 18 (14.2) |

**Notes:**
SSI, surgical site infection.
[*] Included one isolate from January 2020.

antibiotic. The antibiotic susceptibility phenotypes of the ESBL-EC were also tabulated in Table S2.

## ESBL genes profile and genotypic distribution of the ESBL-EC

Overall, the most prevalent gene was $bla_{CTX-M-1}$, whereby the isolates were either harbouring $bla_{CTX-M-1}$ only (22%) or co-habouring $bla_{CTX-M-1}$ and $bla_{TEM}$ (21.3%), $bla_{CTX-M-1}$ and $bla_{SHV}$ (7.1%), as well as all three ESBL genes tested (7.9%). The $bla_{FOX}$ gene was absent in all strains tested. No discernible patterns of increased resistance profile to other beta-lactams including carbapenems, were detected among the isolates carrying the three ESBL genes. Among the flomoxef-resistant strains ($n = 8$), it was observed that each of them harboured one to two ESBL genes; however, none of these strains exhibited elevated ESBL genotypic profile as well as the phenotypic resistant patterns to other beta-lactams. It was observed that the ESBL genes distribution showed variable regional patterns or rather hospital-specific trends (Table 3). Among all $bla_{SHV}$ carriers ($n = 26$), 73.1% ($n = 19$) were isolated from the northeasters, HUSM, while the rest of the isolates

**Table 2 Antibiotic susceptibility phenotypes of 127 ESBL-EC isolates.**

| Antimicrobial agent | MIC range (µg/mL) | MIC$_{50}$ (µg/mL) | MIC$_{90}$ (µg/mL) | Susceptibility phenotype[a] (n) (%) | | |
|---|---|---|---|---|---|---|
| | | | | S | I | R |
| (A) Disk diffusion test | | | | | | |
| Cefpodoxime (10 µg) | N/A | N/A | N/A | 1 (0.8) | 0 (0) | 126 (99.2) |
| Ceftriaxone (30 µg) | N/A | N/A | N/A | 1 (0.8) | 0 (0) | 126 (99.2) |
| Ciprofloxacin (5 µg) | N/A | N/A | N/A | 43 (33.9) | 11 (8.7) | 73 (57.5) |
| Ertapenem (10 µg) | N/A | N/A | N/A | 127 (100) | 0 (0) | 0 (0) |
| (B) Broth microdilution test | | | | | | |
| Amoxicillin-clavulanic acid | 4–>256 | 32 | 64 | 33 (26) | 30 (23.6) | 64 (50.4) |
| Piperacillin-tazobactam | 16–512 | >512 | >512 | 6 (5) | 23 (19) | 92 (76) |
| Cefoxitin | 4–>256 | 16 | 64 | 56 (44.1) | 35 (27.6) | 36 (28.3) |
| Ceftazidime | 2–>256 | 256 | >256 | 10 (7.9) | 5 (3.9) | 112 (88.2) |
| Cefotaxime | 256–>256 | >256 | >256 | 0 (0) | 0 (0) | 127 (100) |
| Cefepime | 32–>256 | >256 | >256 | 0 (0) | 0 (0) | 127 (100) |
| Imipenem | ≤0.25–4 | 0.5 | 2 | 104 (81.9) | 17 (13.4) | 6 (4.7) |
| Meropenem | ≤0.25–2 | ≤0.25 | 0.5 | 124 (97.6) | 3 (2.4) | 0 (0) |
| Aztreonam | 128–>256 | 256 | >256 | 0 (0) | 0 (0) | 127 (100) |
| Trimethoprim-sulfamethoxazole | >64:1,216 | >64:1,216 | >64:1,216 | 0 (0) | 0 (0) | 127 (100) |
| Flomoxef[b] | 0.125–>64 | 4 | 32 | 104 (81.9) | 15 (11.8) | 8 (6.3) |

Notes:
N/A, not applicable.
[a] S, susceptible; I, intermediate; R, resistant.
[b] MIC breakpoints for moxalactam was used as reference (CLSI).

**Table 3 ESBL genes profiles of the ESBL-EC.** ESBL genes profiles of the ESBL-EC from different study sites. Data are presented as n (%).

| ESBL gene profiles | HK-IMU | HS | HCTM UKM | UMMC | HUSM | Total |
|---|---|---|---|---|---|---|
| $bla_{CTX-M-1}$, $bla_{TEM}$, $bla_{SHV}$ | 0 (0) | 0 (0) | 1 (3.1) | 0 (0) | 9 (22.5) | 10 (7.9) |
| $bla_{CTX-M-1}$, $bla_{SHV}$ | 0 (0) | 0 (0) | 0 (0) | 2 (4.9) | 7 (17.5) | 9 (7.1) |
| $bla_{CTX-M-1}$, $bla_{TEM}$ | 5 (71.4) | 2 (28.6) | 3 (9.4) | 10 (24.4) | 7 (17.5) | 27 (21.3) |
| $bla_{TEM}$, $bla_{SHV}$ | 0 (0) | 0 (0) | 1 (3.1) | 2 (4.9) | 0 (0) | 3 (2.4) |
| $bla_{CTX-M-1}$ | 2 (28.6) | 0 (0) | 10 (31.3) | 10 (24.4) | 6 (15.0) | 28 (22.0) |
| $bla_{SHV}$ | 0 (0) | 0 (0) | 0 (0) | 1 (2.4) | 3 (7.5) | 4 (3.1) |
| $bla_{TEM}$ | 0 (0) | 4 (57.1) | 3 (9.4) | 12 (29.3) | 4 (10) | 23 (18.1) |
| None detected | 0 (0) | 1 (14.3) | 14 (43.8) | 4 (9.8) | 4 (10) | 23 (18.1) |
| Total | 7 | 7 | 32 | 41 | 40 | 127 |

Note:
HK-IMU, Hospital Kluang-International Medical University; HS, Hospital Serdang; HCTM UKM, Hospital Canselor Tuanku Muhriz University Kebangsaan Malaysia; UMMC, University Malaya Medical Centre; HUSM, Hospital Universiti Sains Malaysia.

(n = 7, 26.9%) were from the central region. None of the ESBL-EC harboured any carbapenemase genes tested. The genotypic distribution of ESBL-EC was provided in Table S2.

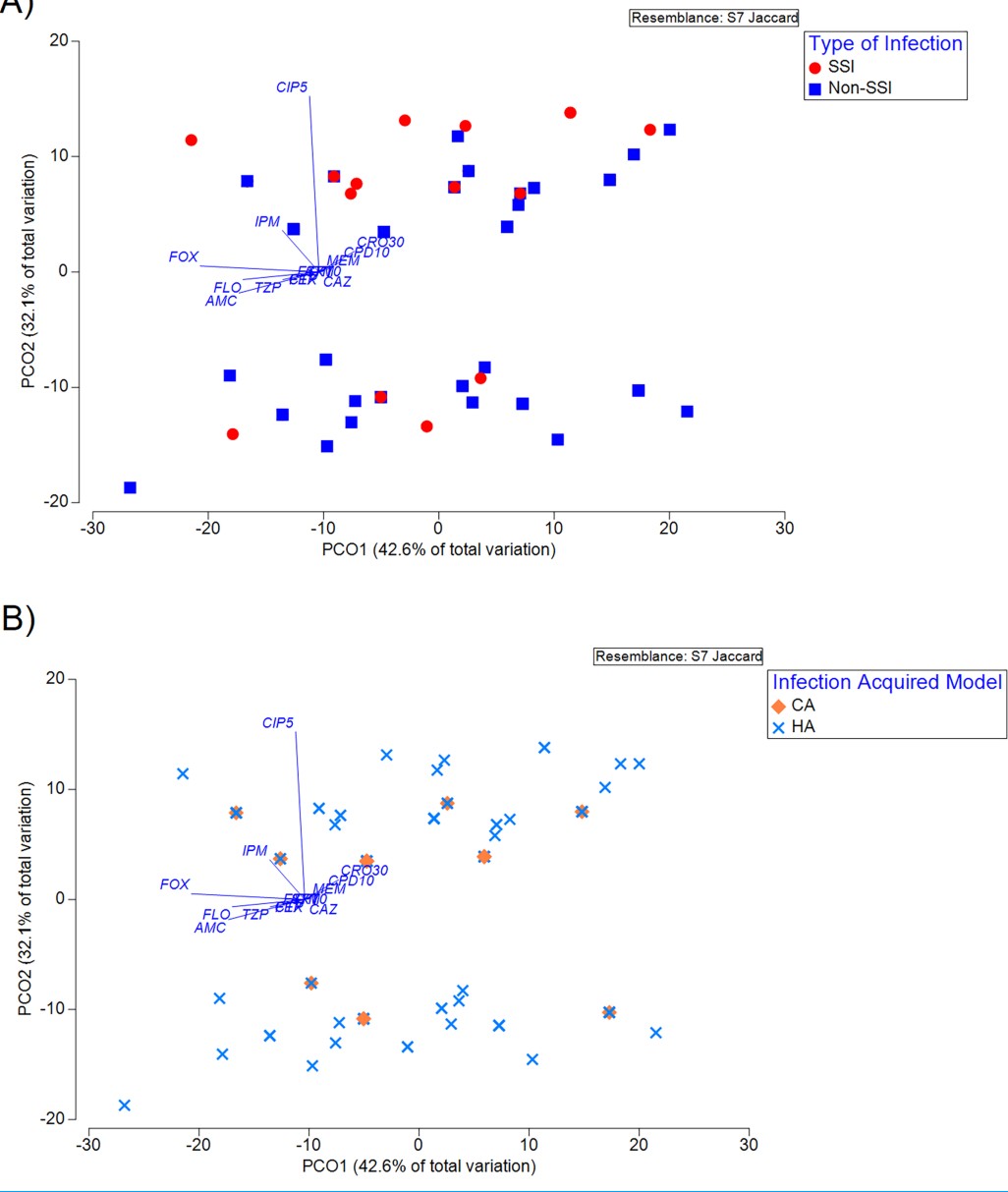

**Figure 1 Principal coordinate analysis (PCoA) of susceptibility phenotype of the isolates classified according to (A) type of infection (TOI), and (B) infection acquired model (IAM).** Principal coordinate analysis (PCoA) of susceptibility phenotype of the isolates classified according to (A) type of infection (TOI), and (B) infection acquired model (IAM) ($R^2$ = 74.7%). The vector overlaid indicates the direction across the ordination plane in which values of the antibiotics increase (resistance). The length of the vector indicates the amount of total variation in each variable is explained in the chosen ordination plane.                                     

## Association of TOI/IAM with demographic factors

The antibiotic susceptibility phenotype profile of the isolates visualised using PCoA showed that no significant clustering across TOI (Fig. 1A) and IAM (Fig. 1B) was observed. As shown in Fig. 1, the associations between susceptibility phenotype and TOI or IAM did not impact the ordination on the PCoA plot, suggesting that these were not the major

**Table 4 Chi-square test results related to the correlation between the demographic factors and TOI or IAM.**

| | Pearson's Chi-squared test | | | |
| | Type of infection (TOI) | | Infection acquired model (IAM) | |
| | X-squared (df) | p-value | X-squared (df) | p-value |
|---|---|---|---|---|
| Gender | 8.554 (8) | 0.3813 | 5.231 (4) | 0.2644 |
| Ethnicity | 106 (8) | <2.2e−16** | 13.143 (4) | 0.0106* |

Notes:
* $p \leq 0.05$.
** $p \leq 0.001$.

**Table 5 GLM results evaluating for the association between antibiotic susceptibilities of the strains with TOI or IAM.**

| | Generalized linear model (GLM) | | | | | | | |
| | Type of infection (TOI) | | | | Infection acquired model (IAM) | | | |
| Antibiotics | Estimate | Std. Error | z value | Pr (>|z|) | Estimate | Std. Error | z value | Pr (>|z|) |
|---|---|---|---|---|---|---|---|---|
| CPD10 | 19.43 | 516.05 | 0.04 | 0.97 | −16.63 | 836.09 | −0.02 | 0.98 |
| CRO30 | 35.64 | 12,692.62 | 0 | 1 | −33.79 | 6,711.99 | −0.01 | 1 |
| ETP10 | RD | RD | RD | RD | RD | RD | RD | RD |
| CIP5 | 0.49 | 0.48 | 1.02 | 0.31 | −1.07 | 0.68 | −1.57 | 0.12 |
| AMC | −0.13 | 0.6 | −0.21 | 0.83 | −2.18 | 1 | −2.18 | 0.03* |
| TZP | 0.81 | 1.01 | 0.8 | 0.42 | −0.04 | 1.38 | −0.03 | 0.97 |
| FOX | 0.06 | 0.54 | 0.12 | 0.91 | −0.32 | 0.68 | −0.48 | 0.63 |
| CAZ | −1.42 | 0.91 | −1.57 | 0.12 | −1.33 | 1.7 | −0.79 | 0.43 |
| CTX | RD | RD | RD | RD | RD | RD | RD | RD |
| FEP | RD | RD | RD | RD | RD | RD | RD | RD |
| IPM | 1.86 | 0.85 | 2.18 | 0.03* | 0.85 | 1.24 | 0.69 | 0.49 |
| MEM | −35.18 | 8,010.3 | 0 | 1 | 30.83 | 1,024 | 0.03 | 0.98 |
| ATM | RD | RD | RD | RD | RD | RD | RD | RD |
| SXT | RD | RD | RD | RD | RD | RD | RD | RD |
| FLO | 0.7 | 0.72 | 0.98 | 0.33 | 0.7 | 0.99 | 0.7 | 0.48 |

Notes:
CPD10, Cefpodoxime; CRO30, Ceftriaxone; ETP10, Ertapenem; CIP5, Ciprofloxacin; AMC, Amoxicillin-clavulanic acid; TZP, Piperacillin-tazobactam; FOX, Cefoxitin; CAZ, Ceftazidime; CTX, Cefotaxime; FEP, Cefepime; IPM, Imipenem; MEM, Meropenem; ATM, Aztreonam; SXT, Trimethoprim-sulfamethoxazole; FLO, Flomoxef.
* Significance at $p \leq 0.05$.

confounders of the observed distribution. Subsequent analyses therefore employed a Pearson's Chi-squared test to evaluate the association of TOI/IAM with gender and ethnicity. It was shown that the ethnicity is a significant confounder for both TOI ($p < 2.2e−16$) and IAM ($p = 0.0106$) (Table 4). Using the GLM regression analysis, we identified significant effects of imipenem and amoxicillin-clavulanic acid in response to TOI and IAM, respectively (Table 5).

## DISCUSSION

The antibiotic susceptibility testing using commonly prescribed antibiotics was performed to determine the susceptibility patterns of the ESBL-EC collected. Our analysis of 127 ESBL-EC from five hospitals throughout peninsular Malaysia showed that the isolates were generally highly resistant to third- and fourth-generation cephalosporins, beta-lactam combination agents, monobactam and sulphonamides. Of note, subpar effectiveness of beta-lactam combination agents was also observed in other study investigating ESBL-EC associated with urinary tract infection (*Kettani Halabi et al., 2021*). High level of resistance to trimethoprim-sulfamethoxazole was also commonly observed among ESBL-EC as reported by other studies (*Kassakian & Mermel, 2014*; *Naziri et al., 2020*), and should be ruled out as therapy options for the treatment of urinary tract infections caused by the ESBL-producers (*Auer, Wojna & Hell, 2010*). Besides carbapenems, flomoxef showed a high susceptibility rate (81.9%) against ESBL-EC. Other published studies including an *in vitro* flomoxef study by *Jung et al. (2019)*, have been focusing on ESBL-producing isolates and the isolates resistant to imipenem have been excluded. In the current study, 18.1% of the isolates were found resistant to imipenem but did not harbour any carbapenemase genes tested. The antibacterial activity of flomoxef against these isolates was not significant as expected; however, flomoxef was active against some of the ESBL-EC with reduced imipenem susceptibility. Collectively, our data reinforces the idea that flomoxef is not inferior to carbapenems *in vitro* based on the breakpoints interpretation. The finding is coherent with a previous investigation focused on ESBL-producers isolated from urinary tract infections, which was performed in one of the five hospitals in the current study (*Ngoi et al., 2021*). The data obtained may open door to future appropriate trials to evaluate the potential clinical efficacy of this drug.

Current data covered three major ethnicities in Malaysia, namely the Malays, Chinese and Indians. Our result suggested ethnicity is the confounding factor that drives infections categorised based on TOI and IAM. Gender has long been identified as a potential confounder of infections with AMR pathogens, but studies detected ethnicity differences are less reported (*Brandl et al., 2021*). Ethnicity can be a complex and multifaceted factor and it is important to consider other potential variables such as genetic factors (*Jenks et al., 2023*), co-morbidities and dietary factors (*Tan, Dunn & Yen, 2011*) *etc*., making it challenging to determine a direct associated relationship. Based on the findings from the present study, future research endeavours may prioritise improvement in data collection approach to capture ethnicity and other potential factors. Additionally, analysing large healthcare databases will be insightful to discern patterns and disparities in infection rates, treatment outcomes and AMR across diverse ethnic groups, holding promise for informing tailored treatment strategies in the future.

In this study, episodes of community-acquired infection accounted for 14.2% of the cases suggesting the changing epidemiology of infections due to the ESBL-producers. This notion is supported by *Dwiyanto et al. (2020)* who reported a high prevalence of intestinal carriage of ESBL-(21.6%) and pAmpC-(32.1%) producing *Enterobacteriaceae* in a healthy sub-urban community in Malaysia (*Dwiyanto et al., 2020*). Similarly, the Malaysian

Tricycle Project initiated by the World Health Organisation (WHO) reported a shocking 20% of ESBL-EC carriage among healthy pregnant women. Furthermore, the findings also demonstrated that ESBL-EC are widely circulating in humans, the food chain and the environment, and the sequenced isolates were all phylogenetically related (*World Health Organization (WHO), 2019*). Additionally, the distribution of ESBL-producing *Enterobacterales* in the water bodies such as hospital wastewater and drinking water has been reported in other parts of the world (*Fadare & Okoh, 2021*; *Mahmud et al., 2020*). This suggests that human medicine may not be the only fundamental driver of the emergence of ESBL-EC. Therefore, the selection and transmission of AMR are multifaceted and complex; preventing and reducing the burden of AMR should be addressed with a One Health approach. A major knowledge gap in the application of the One Health concept is the lack of data in quantifying the magnitudes of AMR selection and transmission factors that promote the currently observed growing incidence of ESBL infections. Quantification could aid the prioritisation of risk reduction strategies and refine policy approaches to untangle the inherently complex challenges of AMR.

Early reports of ESBL-producers typically described SHV and TEM as responsible for serious hospital-acquired infections, and the rise of CTX-M has been characterised by community-onset infections with no direct healthcare contact (*Pitout & Laupland, 2008*). Today CTX-M family enzymes are the most predominant ESBLs worldwide (*Castanheira, Simner & Bradford, 2021*). The current study reported that majority of the isolates either harboured CTX-M-1 alone (22%) or co-harbouring both CTX-M-1 and TEM (21.3%). In this study, we did not observe any trends in ESBL genotypic profile among the flomoxef-resistant strains, although previous study reported that AmpC beta-lactamase production was the prevalent resistance mechanism for flomoxef resistance (*Matsumura et al., 2016*). Nonetheless, the study did not conduct additional molecular testing to definitively determine the specific resistance mechanisms responsible for conferring flomoxef resistance. SHV is the least common ESBL genes detected in the current surveillance study, and we observed that the related genotypes were only confined to HUSM and UMMC. This could be due to the different antimicrobial regimes used in the hospitals which may contribute to the direct or indirect selection for resistance (*Perry et al., 2021*). In our previous single centre study, none of the ESBL-EC tested harboured $bla_{SHV}$ (*Ngoi et al., 2021*). This could explain the reason why the resistance rate of cefoxitin was seen increased (28.3%) in the current study as compared to our previous single centre study (12%) because the TEM- and/or SHV-carriers are ineffective against cephamycins (*Jacoby & Medeiros, 1991*). Additionally, 23 isolates (18.1%) in the current study did not carry any of the four ESBL genes tested, thus future studies should include other ESBL genes such as other CTX-M-ases, $bla_{OXA}$, $bla_{PER}$, $bla_{GES}$, *etc.*, in order to gain a more comprehensive view of molecular epidemiology of ESBL-EC in Malaysia.

This study has its strengths and limitations. The isolates and data collected are not risk-adjusted. Another limitation of this study was the inconsistency of sample collection, in which an equal number of isolates should have been collected throughout each centre. The sample collection in each centre was performed through convenient sampling due to resource constraints. Hence, comparison and monitoring trends between centres and years

were not possible. Molecular epidemiology data of ESBL-EC causing community and hospital-acquired infections in Malaysia is insufficient. Thus current hospital-based multicentre study provided an updated overview of the distribution and molecular characterisation of the ESBL-EC in Malaysia. Data collected from this study supports the ongoing monitoring trends of AMR at local, national and international levels.

## CONCLUSIONS

This study demonstrated that ESBL-EC strains isolated from five healthcare centres in Malaysia were highly resistant to third- and fourth-generation cephalosporins, beta-lactam combination agents, monobactam, fluoroquinolone and sulphonamides. Carbapenems exhibited the highest *in vitro* efficacy against ESBL-EC, followed by flomoxef. Collectively, the findings provide a rationale for further clinical studies assessing flomoxef as an alternative to carbapenems in Malaysia for treating infections caused by flomoxef-susceptible ESBL-EC. There is an urgent need for interventions, including regular surveillance of ESBL-producers and antibiotic rotation at both institutional and national levels to maintain the antimicrobial armamentarium and prevent the development of resistance to carbapenems among the local ESBL-producing *Enterobacterales*.

## ACKNOWLEDGEMENTS

This was a collaborative study between the University of Malaya (UM), National University of Malaysia (UKM), University of Science Malaysia (USM), University of Putra Malaysia (UPM), International Medical University (IMU) and Shionogi Singapore Pte. Ltd. We wish to thank our collaborators, especially UM and Shionogi for providing the research materials and facilities for this study.

### Funding

This research was funded by the Ministry of Higher Education (Malaysia), Transdisciplinary Research Grant Scheme, TRGS (TRGS/1/2020/UM/02/2/2) under project code TR001B-2020 and Shionogi Singapore Pte. Ltd (IF015-2018). The funders had no role in study design, data collection and analysis, decision to publish, or preparation of the manuscript.

### Grant Disclosures

The following grant information was disclosed by the authors:
Ministry of Higher Education (Malaysia), Transdisciplinary Research Grant Scheme, TRGS (TRGS/1/2020/UM/02/2/2): TR001B-2020.
Shionogi Singapore Pte. Ltd: IF015-2018.

### Competing Interests

Cindy Shuan Ju Teh is an Academic Editor for PeerJ. Loong Hua Tee and Kin Chong Leong were employed by Shionogi Singapore Pte Ltd at the time of the study.

## Author Contributions

- Polly Soo Xi Yap performed the experiments, analyzed the data, prepared figures and/or tables, authored or reviewed drafts of the article, and approved the final draft.
- Chun Wie Chong analyzed the data, prepared figures and/or tables, authored or reviewed drafts of the article, and approved the final draft.
- Sasheela Ponnampalavanar conceived and designed the experiments, authored or reviewed drafts of the article, and approved the final draft.
- Ramliza Ramli analyzed the data, authored or reviewed drafts of the article, and approved the final draft.
- Azian Harun analyzed the data, authored or reviewed drafts of the article, and approved the final draft.
- Tengku Zetty Maztura Tengku Jamaluddin analyzed the data, authored or reviewed drafts of the article, and approved the final draft.
- Anis Ahmed Khan analyzed the data, authored or reviewed drafts of the article, and approved the final draft.
- Soo Tein Ngoi performed the experiments, authored or reviewed drafts of the article, and approved the final draft.
- Yee Qing Lee performed the experiments, authored or reviewed drafts of the article, and approved the final draft.
- Min Yi Lau performed the experiments, authored or reviewed drafts of the article, and approved the final draft.
- Shiang Chiet Tan performed the experiments, authored or reviewed drafts of the article, and approved the final draft.
- Zhi Xian Kong performed the experiments, authored or reviewed drafts of the article, and approved the final draft.
- Jia Jie Woon performed the experiments, authored or reviewed drafts of the article, and approved the final draft.
- Siew Thong Mak performed the experiments, authored or reviewed drafts of the article, and approved the final draft.
- Kartini Abdul Jabar analyzed the data, authored or reviewed drafts of the article, and approved the final draft.
- Rina Karunakaran analyzed the data, prepared figures and/or tables, authored or reviewed drafts of the article, and approved the final draft.
- Zalina Ismail analyzed the data, authored or reviewed drafts of the article, and approved the final draft.
- Sharifah Azura Salleh analyzed the data, authored or reviewed drafts of the article, and approved the final draft.
- Siti Suraiya Md Noor analyzed the data, authored or reviewed drafts of the article, and approved the final draft.
- Siti Norbaya Masri analyzed the data, authored or reviewed drafts of the article, and approved the final draft.

- Niazlin Mohd Taib analyzed the data, authored or reviewed drafts of the article, and approved the final draft.
- Azmiza Syawani Jasni analyzed the data, authored or reviewed drafts of the article, and approved the final draft.
- Loong Hua Tee analyzed the data, authored or reviewed drafts of the article, and approved the final draft.
- Kin Chong Leong analyzed the data, authored or reviewed drafts of the article, and approved the final draft.
- Victor Kok Eow Lim analyzed the data, authored or reviewed drafts of the article, and approved the final draft.
- Sazaly Abu Bakar conceived and designed the experiments, authored or reviewed drafts of the article, and approved the final draft.
- Cindy Shuan Ju Teh conceived and designed the experiments, prepared figures and/or tables, authored or reviewed drafts of the article, and approved the final draft.

### Ethics

The following information was supplied relating to ethical approvals (*i.e.*, approving body and any reference numbers):

The study was conducted according to the guidelines of the Declaration of Helsinki. The study was registered in the Malaysian National Medical Research Register (NMRR) and approved by the Malaysian Medical Research and Ethics Committee (MREC) (NMRR-19-1314-46718 IIR).

### Data Availability

The raw data of the numerical values for MIC are available in the Supplemental File.

### Supplemental Information

Supplemental information for this article can be found online at http://dx.doi.org/10.7717/peerj.16393#supplemental-information.

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
