# Peer review of "A multicentre study to determine the in vitro efficacy of flomoxef against extended-spectrum beta-lactamase producing Escherichia coli in Malaysia"

_PeerJ, doi:10.7717/peerj.16393_

## Round 0.1 · original submission · Minor Revisions

While reviewers showed overall enthusiasm for the work, they also shared comments about the need for more information about the ESBL gene and the strains harboring these genes. In particular, please include insights into the mechanisms driving the resistance and the implications for treatment, as well as explain whether strains with multiple ESBL genes have a resistant phenotype against multiple antibiotics.

Reviewer 1 ·

Basic reporting

The manuscript is well structured and impressive in findings. All tables and figures are clearly presented and support the hypothesis/conclusions throughout the story. Minor revisions may be considered to improve the quality and soundness of this manuscript, which are listed in section 2-4 below.

Experimental design

The authors may consider to briefly describe how the total of 127 ESBL-EC strains were collected at participating hospitals and later confirmed at the central laboratory. This will provide a better reference for future similar studies at other regions in the world.

Validity of the findings

It will be helpful if the authors can discuss more on the potential correlations between ESBL genes identified in Table 3 and the antibiotic resistance phenotype observed in Table 2. For example, flomoxef was shown to have good in vitro activity against ESBL-producing E.coli and K.pneumoniae. For those flomoxef-resistant ESBL-EC strains in this study, are they equipped with ESBL genes and resistant to other antibiotics as well? Similarly, for those strains with multiple ESBL genes, do they have a resistant phenotype against multiple antibiotics?

Additional comments

There is a grammar issue from Line 223 to 224.

Reviewer 2 ·

Basic reporting

This research study presents a deep-dive analysis into the antibiotic susceptibility patterns of extended-spectrum β-lactamase-producing Escherichia coli (ESBL-EC) in peninsular Malaysia. ESBL is an ever-concerning issue, particularly for developing and low income countries. The study does commendably in identifying the resistance patterns to various classes of antibiotics and highlighting the potential efficacy of flomoxef against ESBL-EC, which could indeed pave the way for future clinical trials and could be used as a cost-effective alternative treatment. Moreover, the paper recognizes the multidimensional nature of antibiotic resistance development, emphasizing a 'One Health' approach that considers the interconnectedness of humans, animals, and the environment in tackling this global health threat.

Experimental design

No comment

Validity of the findings

The effort to link the occurrence of antibiotic-resistant infections with ethnicity and gender also adds a unique angle to the study, potentially uncovering novel determinants of infection rates. However, the study seems to fall short in substantiating the claims regarding the influence of ethnicity on antibiotic resistance. While references are made to other works that discuss variations in gut microbiota amongst different ethnic groups, there is a lack of direct evidence within the study to corroborate the suggested link between ethnicity and susceptibility to ESBL-EC infections. This angle, though interesting, requires a more rigorous analysis to avoid potential oversimplification and misinterpretation.

In terms of the molecular epidemiology section, while the paper does well to highlight the prevalence of various ESBL genes in the isolates, it doesn't delve deep enough. It would have been enriching to see a more detailed analysis here, including insights into the mechanisms driving the resistance and the implications for treatment and management of infections caused by these strains. Is it possible to treat different ESBL genes individually to find treatment options or a single treatment option is better for all strains harboring any of the ESBL gene?

Additional comments

No comment

---

## Round 0.2 · accepted · Accept

I confirm that the authors have addressed all of the reviewers' comments, and the current version is ready for publication.

Reviewer 1 ·

Basic reporting

The authors have well addressed the questions and concerns from our end. The story is now in very good quality for publication purpose.

Experimental design

No comment.

Validity of the findings

No comment.

Reviewer 2 ·

Basic reporting

The authors have adequately responded to the comments mentioned in the previous review. The reviewer is endorsing the manuscript for publication.

Experimental design

no comment

Validity of the findings

no comment